# Neutrophil Extracellular Vesicles: A Delicate Balance between Pro-Inflammatory Responses and Anti-Inflammatory Therapies

**DOI:** 10.3390/cells11203318

**Published:** 2022-10-21

**Authors:** Yang Zhou, Sabrina Bréchard

**Affiliations:** Signal Transduction Group, Department of Life Science and Medicine, University of Luxembourg, L-4367 Belvaux, Luxembourg

**Keywords:** neutrophils, inflammation, extracellular vesicles, therapy

## Abstract

Extracellular vesicles (EVs) are released in the extracellular environment during cell activation or apoptosis. Working as signal transducers, EVs are important mediators of intercellular communication through the convoying of proteins, nucleic acids, lipids, and metabolites. Neutrophil extracellular vesicles (nEVs) contain molecules acting as key modulators of inflammation and immune responses. Due to their potential as therapeutic tools, studies about nEVs have been increasing in recent years. However, our knowledge about nEVs is still in its infancy. In this review, we summarize the current understanding of the role of nEVs in the framework of neutrophil inflammation functions and disease development. The therapeutic potential of nEVs as clinical treatment strategies is deeply discussed. Moreover, the promising research landscape of nEVs in the near future is also examined.

## 1. Extracellular Vesicles: What Do We Know?

Although, the existence of extracellular vesicles (EVs) has been suspected since 1946, originally as procoagulant platelet-derived particles [1], they were named much later (1971) following the ultrastructural identification of a large range of intra- and extracellular membranous structures that were recovered by the centrifugation of flagellated algae [2]. 

Meanwhile, these particles were referred to as “platelet dust” [3], a term that reflects the long-lasting belief that EVs were simply inert cellular debris. 

However, over the decades, the interest in these nanovesicles has always been sparked by new discoveries of their roles in disease pathogenesis and their therapeutic potentials. Therefore, the historic view of the role of EVs has evolved considerably over the years; initially considered waste-removal carriers, they are now regarded as major mediators of intercellular communication [4,5,6] by transporting a variety of molecules. 

EVs are released in the extracellular environment during cell activation or apoptosis and are circulating in most human biological fluids and tissues, crossing endothelial barriers to interact with cells in a paracrine or autocrine fashion [7].

These signal transducers express various ligands and receptors emanating from their original cells and serve as cargo vehicles to transport a complex array of information to the surroundings or distant target cells [8] via the delivery of a plethora of biologically active molecules such as lipids, proteins, mRNAs, noncoding RNA, miRNAs, and metabolites [9,10]. Through the release of such selective molecules, EVs can support or even trigger the change of cell phenotypes through epigenetic reprogramming [11] and the adaptation of cellular responses to environmental challenges by modulating crucial processes such as homeostasis, antigen presentation, signal transduction, the modulation of inflammation, immune responses, and the promotion of repair functions [7,12,13]. 

Over the years, the classification included three specific categories of EVs, which were distinguished according to their sizes and origins. One of the categories encompassed small EVs (50–150 nm), which were assimilated under the term “exosomes”. Their formation derives from the invagination of endosomal membranes of internal multivesicular bodies and the fusion of these compartments with the plasma membrane [14,15,16]. A second subtype of vesicles, designated medium EVs (100–1000 nm), is generated by the direct outward budding and shedding of the plasma membrane [17] and has been assigned various other names such as microvesicles, microparticles, or ectosomes. Finally, apoptotic bodies are formed by the plasma membrane blebbing of apoptotic cells [15] and are considered cellular debris with minimal biological functions [18]. 

EV classification is still rather arbitrary and suffers from the fact that these multifunctional structures share common features, with no specific markers to distinguish them. Moreover, the overlapping sizes of ectosomes and exosomes in this nomenclature is rather irrelevant and has led to number confusing scientific publications (for a review, see [19]). In fact, the term “exosomes” took the lead over the other designations and has become the most popular and generic word to describe EVs. 

To overcome this problem, in their last guidelines, the International Society for Extracellular Vesicles (ISEV2018) [20] recommended to use EVs as the consensus term to identify “heterogeneous membrane-bound vesicles limited by phospholipid bilayers with the incapacity to replicate due to lack of a functional nucleus”. An operational term to distinguish EV subtypes should refer to size/density, biochemical composition, and descriptions of the conditions of the cell of origin. The specific terms “exosomes” and “ectosomes” should be exclusively used to identify multivesicular-body-derived EVs and plasma-membrane-derived EVs, respectively, regardless of their sizes [20]. Ideally, EVs should eventually be classified by different protein profiles due to their different methods of formation, but this is still not feasible. Indeed, some proteins, abundant in one vesicle, were later shown to also be present in other vesicle types, sometimes in different amounts [20].

Beyond moderate differences in EV physical properties, the current bottleneck in the understanding of the distinct functional effects of EV subtypes is the lack of reliable markers. Several tetraspanins, notably CD63, CD81, and CD9 have been used as markers of exosomes for the last two decades, but more recently their presence in other EVs has been observed [21]. However, it is conceivable that EVs bearing only CD9 or CD81 but not CD63 could correspond to ectosomes, while those expressing CD63 in combination with CD9 and/or CD81 could be related to late endosomal-derived vesicles [21]. Substantial validation is required to discriminate exosomes from small ectosomes in any cell type and to achieve harmonization for EV isolation and analysis methods to prevent a nonspecific overlap of biomarkers that may disguise the exact nature of EVs. Understanding the specific functional properties of each subpopulation more distinctly and profoundly constitutes a basis to fully exploit the enormous potential of EVs in terms of therapeutic targets and circulating biomarkers. This need is urgent, given the recent discovery of another class of particle, the exomeres. 

Nonmembranous extracellular nanovesicles, named exomeres, are small (approximately 35 nm) and were initially isolated using the separation technique of asymmetric-flow field-flow fractionation [22]. Thereafter, they were isolated by an optimized ultracentrifugation method [23]. Although exomeres can carry nucleic acids, proteins, and lipids, which can be delivered to recipient cells, the absence of a lipid bilayer and a biogenesis distinct from the plasma membrane and the endocytic pathway argue for the fact that exomeres should not be classified as EVs but rather assigned to a new type of extracellular particle [24]. Despite such differences, exomeres possess functional activities that are clinically relevant. In a recent and very exciting article published in Cell Reports, Zhang and coworkers [23] highlighted that exomeres can have a pro-neoplastic role in cancer by transferring cargo proteins with functional activities such as β-galactoside α2,6-sialyltransferase 1 and amphiregulin, which participate in cell proliferation and affect the invasiveness of cancer cells [25,26,27,28]. 

In another study, the supernatant of ultracentrifugation was used to retrieve exomeres, which were subjected to another step of centrifugation, and that led to the identification of yet another type of extracellular particle, named “supermeres” [29]. Supermeres can induce phenotypic and metabolic changes in recipient cells as well as the transfer of drug resistance. It is worth noting that the RNA and protein composition of supermeres differs from that of exomeres and small extracellular vesicles. Supermeres harbor multiple cargos that are related to a large panel of pathologies, including Alzheimer’s (e.g., amyloid precursor protein, APP) and cardiovascular (e.g., angiotensin-converting enzyme, ACE) diseases as well as cancer (e.g., MET proto-oncogene and receptor tyrosine kinase). In addition, supermeres contain high quantities of extracellular RNA, much higher than the amounts found in small extracellular vesicles and exomeres [29], with a strong enrichment of miR-1246, which is considered an oncogenic miRNA in a variety of cancer types [30,31,32,33]. 

## 2. Extracellular Vesicles of Neutrophils (nEVs)

Neutrophils are known to exert their functions through the release of a large panel of cytotoxic enzymes and effector molecules, phagocytosis, the production of reactive oxygen species (ROS), and the formation of neutrophil extracellular traps (NETs) (Figure 1). 

While data on the EVs released from immune cells are in constant expansion, much less is known about the role of EVs derived from neutrophils (nEVs). nEVs are not fully explored yet, probably because of the short life span of neutrophils and the difficulties in isolating and manipulating them. However, given the immunogenic and proinflammatory roles of EVs, it is now well-established that nEVs can act as an essential element between neutrophil-driven inflammation and tissue damage [47,48,49,50,51].

nEVs were first identified by Stein and Luzio [52], and over the years a consensus has emerged that neutrophils predominantly release medium-sized vesicles (ectosomes) and, to a smaller extent, small-sized vesicles (exosomes). Commonly, nEVs express CD11b, CD18, CD66b, and myeloperoxidase (MPO) on their surface and can be stained with annexin V, indicating that phosphatidylserine is exposed on the outer leaflet of nEVs [53,54]. Although nEVs can differ in terms of protein composition and functional properties, they share similar physical and chemical characteristics such as size and surface properties [53,54,55,56,57]. nEVs contain a large amount of granule proteins known for their antibacterial activity. Upon neutrophil stimulation with opsonized particles, bacterial growth was reduced through the release of nEVs enriched with granules. In the same study, the authors showed that nEVs were also able to mediate bacterial aggregation, which relied on an integrin-dependent interaction of nEVs with bacteria and involved the actin cytoskeleton [54]. It was hypothesized that a bacterial aggregate could be exposed to granule proteins packaged in nEVs, which resulted in an inhibition of bacterial growth. Moreover, the quantity of nEVs was increased in the serum of patients with bacteremia, arguing for the in vivo antibacterial effects of nEVs. 

In addition of their antibacterial role, nEVs have been reported to show dual pro- and anti-inflammatory features according to the types of stimuli to which neutrophils are subjected. The proinflammatory functions of nEVs have been associated with an increase in ROS production and cytokine secretion in an autocrine/paracrine manner [57]. The anti-inflammatory effect of nEVs has been shown on immature monocyte-derived dendritic cells and macrophages. nEVs were able to decrease phagocytic activity and increase TGF-β1 release by dendritic cells. When these cells were stimulated by lipopolysaccharide (LPS), nEVs triggered a reduction not only in phagocytic activity but also a decrease in maturation, cytokine release, and T-cell proliferation [58]. Similarly, nEVs increased the release of TGF-β1 by zymosan or LPS-stimulated macrophages [59]. It is likely that the phosphatidylserine on the nEV surface participates in the blocking of the inflammatory response by facilitating the binding of nEVs on cells [60].

In the recent literature, it was proposed that nEVs can be distinguished and separated on the basis of their production into two subtypes that share similar characteristics and can be generated either spontaneously or in response to diverse immunological stimuli (e.g., bacterial stimulation, antibodies, inflammatory and immunosuppressive cytokines, chemokines, and complement components) [61,62,63,64].

On one hand, neutrophil-derived microvesicles (NDMVs) are released following the activation of neutrophils, which are derived from inflammatory foci (Figure 1) [63,64]. The term “microvesicles” in NDMVs is confusing since microvesicles can be designed by ectosomes in the literature and NDMVs could also include exosomes.

On the other hand, neutrophil-derived trails (NDTRs) are generated by neutrophils migrating from blood vessels to sites of inflammation (Figure 1) [63,64]. During migration through inflamed tissue, the neutrophil uropods are elongated and the vascular wall is subjected to a variety of mechanical forces, allowing for the detachment of the tail portion and ultimately the release of mediator-containing EVs [65,66]. Through NDTRs, neutrophils can transmit inflammatory signals to neighboring cells that can eradicate or disseminate inflammation by facilitating neutrophil diapedesis and the subsequent orchestration of the recruitment and activation of other immune cells. 

NDTRs have been assigned proinflammatory roles, and NDMVs have anti-inflammatory functions [63,64]. However, this antagonism might be a premature shortcut simplification of the roles of these two types of nEVs. Indeed, the effects of nEVs are probably largely influenced by the activation signal acting on neutrophils. In other words, on the same cell, nEVs could have opposing effects depending on the neutrophil activation state, which determines the set of active molecules packed into the nEVs and thus the functional effects on the target cells. In an elegant study, Kolonics and coworkers [51] provided convincing data on the ability of resting or apoptotic neutrophils to produce EVs with an anti-inflammatory capacity on neighboring cells. EVs from resting neutrophils were able to decrease the level of ROS and IL-8 secretion from neutrophils isolated from venous blood. Clearly, EVs from apoptotic neutrophils had a procoagulant effect, highlighting their potential involvement in the resolution of inflammation. In contrast, EVs produced by neutrophils activated through opsonized particles led to an increase in ROS and IL-8 secretion from neutrophils and endothelial cells, resulting in antibacterial and proinflammatory activities [51,54].

In addition, we need to keep in mind that distinct biological roles can be attributed to nEVs according to (i) the nature and identity of the inflammatory mediators that have induced neutrophil priming/activation and (ii) the targeted recipient cells. Thus, contrary to the previously observed results, EVs derived from neutrophils activated by N-formylmethionyl-leucyl-phenylalanine (fMLF) or the complement fragment C5a were able to exert an anti-inflammatory/immunosuppressive effect by inhibiting the release of IL-8, IL-10, and TNFα in activated macrophages but conversely promoted TGF-β release [53]. The inhibition of cytokine secretion is presumed to be caused by the activation of the PI3K/Akt pathway mediated by MerTK (Mer proto-oncogene tyrosine kinase), resulting in the inhibition of NF-κB *p*65 phosphorylation and NF-κB translocation [47], whereas the release of TGF-β could be associated with Ca^2+^ influx and was independent of MerTK signaling [60]. 

These divergent activities of nEVs can be explained by the differences between their contents and the differential abundance of proteins. For example, nEVs produced after stimulation by opsonized particles contain a higher quantity of proteins associated with cell adhesion and the immune response and, to a lesser extent, proteins related to the MAPK signaling cascade than EVs derived from apoptotic or resting neutrophils [51,57]. Interestingly, the activation of endothelial cells, leading to the adhesion of immune cells can occur with the inhibition of monocyte/macrophage activation by nEVs [61,67]. 

## 3. nEVs in Diseases

Neutrophils are able to impact all cell types in their environment and can orchestrate adaptive immune responses via the production of EVs, which adapt their functional capacities under the influence of environmental factors. 

EVs are closely related to their source cells, and thus EVs derived from neutrophils carry the specific functions of these cells. The clinical manifestation of this, since neutrophils are the first cells to be present at the site of inflammation, with the initial role to trigger an inflammatory response to eliminate pathogens, is the participation of nEVs in tissue destruction and in the pathomechanisms of a range of various diseases (autoimmune diseases, allergies, and chronic stable inflammatory diseases) by changing the behaviors and functions of target cells. nEVs can also disturb the microenvironment and suppress immune functions in a paracrine and autocrine mode, contributing to the development of cancer [68]. 

One of the best-characterized roles of nEVs is based on the interaction between platelets and circulating neutrophils. The generation of nEVs is mediated by platelet glycoprotein Ibα (GPIbα) after the adhesion of activated platelets to intravascular neutrophils via P-selectin, which interacts with its ligand, P-selectin glycoprotein ligand-1 (PSGL-1), expressed on neutrophils. The content of these nEVs is especially abundant in arachidonic acid, which is shuttled from neutrophils to platelets and is internalized by platelets through the involvement of clathrin and Mac1. The intracellular compartment is enriched in cyclooxygenase 1 (COX-1), which is required to convert arachidonic acid into thromboxane A2. In turn, thromboxane A2 increases the expression of endothelial intercellular adhesion molecule-1 (ICAM-1) and fosters neutrophil functions by facilitating their extravasation through the enhancement of rolling, crawling, and transmigration [69]. In addition, COX1-deficient mice have been shown to be deficient in neutrophil recruitment and pathogen clearance during bacterial pneumonia, underlining the role of COX-1 in innate immunity [69]. 

The role of nEVs as critical mediators of pathologic signaling has also been shown in airway disorders. EVs released by neutrophils are internalized by airway smooth muscle cells, triggering an increase in their proliferative properties and a modulation of apoptosis with, as a consequence, a distortion in airway architecture and responses creating a favorable environment for the exacerbation of inflammation in severe asthma patients and corticosteroid-insensitive asthmatics [70]. 

nEVs are also involved in other airway diseases associated with neutrophil-driven chronic inflammation. A study by the Blalock group [71] excellently demonstrated the importance of neutrophil elastase (NE) activity derived from nEVs in chronic obstructive pulmonary disease (COPD) [71]. nEVs from activated neutrophils harbor NE in a catalytically active orientation on their surface, obstructing the binding of α1-antitrypsine and its protease inhibitor activity. 

Furthermore, nEVs are able to physically associate with collagen fibrils by CD11b/CD18 via the αM-I domain. NE is therefore able to evade antiprotease inactivation [72] and can degrade collagen fibrils and elastin with impunity, resulting in the promotion of the proteolytic destruction of lung extracellular matrix (ECM) and epithelial cell injury, causing emphysema [73]. In addition to provoking a COPD-like phenotype in a murine intratracheal transfer model when infused into the lungs of mice, nEVs derived from activated neutrophils purified from the bronchoalveolar lavage fluid (BALF) of human COPD patients were able to confer a COPD-like phenotype from human to mice in an NE-dependent manner [71,73]. 

As previously shown at the intestinal level [74], it is highly possible that MPO from nEVs participates in airway epithelial damage by causing the loss of epithelial cadherin and thus the modification of extracellular matrix material. MPO from nEVs could also delay wound healing by inhibiting epithelial cell spreading and migration by interfering with actin dynamics.

Moreover, the disruption of intercellular junctions between endothelial cells could promote neutrophil transepithelial migration [75] through high levels of active nEV-derived MMP-9, which could be involved in the cleavage of E-cadherin and desmosomal cadherin family members [75] present in the adherens junctions and allowing cell–cell adhesion in the epithelium [76] (Figure 2). 

The importance of nEVs in COPD was confirmed in a clinical study conducted by Soni and coworkers [77], who correlated the nEVs present in BALF with index scores such as airway obstruction, hyperinflation, gas transfer, exercise tolerance, and dyspnea associated with COPD disease severity [77]. 

Another type of protease has been involved in a feed-forward inflammatory process in cystic fibrosis (CF) airway disease by sustaining inflammasome activation. Indeed, caspase-1 issued from EVs derived from neutrophils conditioned by the CF airway milieu can not only induce increases in the extracellular inflammatory mediators IL-1α, IL-1β, and IL-18 and ICAM-1 in resident epithelial cells but can also induce inflammasome signaling in naïve neutrophils freshly recruited from the blood [78].

The critical role of nEVs on naïve neutrophils may explain how acute local inflammation in a limited area can expand and become chronic. This point is illustrated by the recent work of Nauseef’s group [79], which demonstrated that the treatment of naïve neutrophils with EVs derived from fMLF-stimulated neutrophils primed NADPH oxidase activity. This is reflected by an increase in ROS production in response to a suboptimal concentration of fMLF. The primed oxidase activity was associated with a redistribution of flavocytochrome b_558_ to the cell surface by the fusion of secretory vesicles and specific granules with the plasma membrane and the phosphorylation of SER345 on p47*^phox^*, a cytosolic subunit of NADPH oxidase [79]. Compelling evidence supports the fact that an aberrant level of ROS is involved in the destruction of healthy tissue by damaging biomolecules including DNA, lipids, carbohydrates, and proteins. For example, an imbalance between ROS production and antioxidant systems can disrupt histone deacetylase activity, facilitating the binding of transcription factors to specific DNA motifs, leading to an increase in inflammatory gene expression [80,81].

In general, the failure of the endothelium, provoked by the uptake of nEVs, facilitates neutrophil recruitment and creates the amplification of an inflammatory loop. It would be reductive to confine the role of nEVs to the direct degradation of tissue by the release of a large panel of cytotoxic molecules and proteolytic enzymes. 

Among the nEV-derived proteins contributing to a perpetuating inflammatory state, S100 Ca^2+^-binding protein A9 (S100A9) could be of particular interest since it has been considered a critical endogenous damage-associated molecular pattern protein [82] in relation to its cytokine-like functions and its ability to recruit neighboring cells. Upon the influence of environmental factors, S100A9 can dimerize with S100A8 to form a noncovalently linked protein complexes, which biologically function through the p38 MAPK and ERK1/2 signaling pathways via the involvement of pattern recognition receptors such as Toll-like receptor 4 (TLR4) or receptors for advanced glycation end products (RAGE) [83,84,85] (Figure 2).

S100A8/A9 have been shown to increase the expression of CD11b/CD18 at the rolling neutrophil cell surface and ICAM-1 on endothelial cells, facilitating the interaction between both cell types and thus neutrophil diapedesis [86,87,88]. Closely related to their DAMP properties, S100A8/A9 can efficiently promote the expression and secretion of proinflammatory mediators by diverse cell types [83,89,90,91,92,93], probably via MyD88-dependant TLR4 signaling, resulting in the activation of NF-κB [89,94].

S100A8/A9 have also been described as being involved as key effectors in the pathogenesis of chronic inflammatory diseases [94,95,96,97,98]. For instance, in rheumatoid arthritis (RA), S100A8/A9 has been reported to mediate chondrocyte activation, osteoclast differentiation, leukocyte infiltration in joints, proinflammatory cytokine production by monocytes, and synovial fibroblast proliferation, contributing to cartilage degradation and bone resorption [99,100].

The secretion of S100A8/A9, allowing them to exert extracellular functions, occurs through the formation of NETs [41,101,102] since S100 proteins are known to have no leader sequence and are not transported via the classical endoplasmic reticulum/Golgi pathway [103]. However, S100A8/A9 has also been reported to be present on the surface of EVs [70,104], which could ensure their transfer and secretion to target sites where they can exert their proinflammatory activities. Thus, it is not surprising that S100A8/A9 derived from nEVs are involved in the suppression of the proliferation and migration of human dermal microvascular endothelial cells and the impairment of angiogenesis, a hallmark of systemic sclerosis.

nEVs are also involved in the pathogenesis of other systemic autoimmune disorders, notably antineutrophil cytoplasmic antibody (ANCA)-associated vasculitis (AAV) [105]. AAV is characterized by the inflammation of small vessels, resulting in vascular destruction. The presence of ANCAs directed against proteinase-3 or MPO is one of the characteristics of AAV [106,107]. ANCAs have the potential to activate primed neutrophils, triggering the release of nEVs [61], which have been reported to activate the coagulation cascade [108]. Tissue factor is considered, in vivo, as the primary initiator of the coagulation process and can trigger, upon pathological conditions, arterial and venous thrombosis [109,110,111]. The work of Kambas and coworkers provided evidence that nEVs, through the delivery of tissue factor, participate in inflammation-driven thrombotic diseases. In this sense, nEVs constitute a bridge between inflammation and thrombogenicity in AAV [105].

Besides bioactive proteins, nEVs are also enriched in specific small noncoding RNAs that negatively regulate gene expression via mRNA degradation and/or the translational repression of their targeted mRNAs. The enthusiasm surrounding neutrophil miRNAs for their role in carcinogenesis and autoimmune diseases emerged only a few years ago, but it already echoes at the level of nEVs.

The membrane of EVs can protect miRNAs from degradation by environmental RNases [112,113,114]. Evidence has accumulated that specific miRNAs can be exchanged by nEVs, giving them the potential for immune modulation and to interfere with the functionality of target recipient cells.

For example, miR-30d-5p packed in nEVs released from TNF-α-activated neutrophils was described, in an in vitro coculture model, to target SOCS-1 and SIRT1, negative regulators of the NF-κB signaling pathway [115,116]. This caused an increase in the acetylation of lysine 310 of p65 and the subsequent activation of NF-κB in macrophages, resulting in the induction of macrophage polarization towards a proinflammatory phenotype (M1) and the priming of macrophage cell death by pyroptosis [117], which contributes to postsepsis inflammation and lung tissue injury (Figure 2).

Another miRNA derived from nEVs is involved in the NF-κB signaling pathway in the framework of the chronic inflammation occurring in atherosclerosis. It was observed that circulating nEVs increased during hypercholesterolemia, preferentially adhere to atherosusceptible sites, and deliver miRNA-155 to endothelial cells. miRNA-155 triggers an increase in NF-κB expression by downregulating the transcription repressor BCL6 and the enhancement of inflammation conditions. This vicious circle is exacerbated by the presence of CD18 on the nEV surface, which is responsible for a more intense recruitment of monocytes to the vessel wall, acting as a critical factor in atherosclerotic plaque growth [118]. Recently, a role for miR-142–3p and miR-451 derived from nEVs was highlighted in endothelial damage. These two miRNAs delivered to endothelial cells by nEVs were able to increase apoptosis and the expression of proinflammatory cytokines in these cells and altered angiogenic repair, which can lead to vascular damage [119].

In summary, the functionally diverse roles of miRNAs packed in nEVs depend on the crosstalk between neutrophils, target cells, and the downstream activated molecular networks.

Besides miRNAs, long noncoding RNAs (lncRNAs) have been found in nEVs, and their transfer to recipient cells has been shown to be a source of phenotypic transitions by the epigenetic mechanisms regulating gene expression [120,121,122,123]. It has been reported that CRNDE, affiliated with the lncRNA class, from nEVs activates the NF-κB pathway by enhancing TAK1-mediated IKKβ phosphorylation in airway smooth muscle, thus affecting the proliferation and migration of cells. In support of the role of CRNDE in airway remodeling in asthma, neutrophil CRNDE knockdown in a mouse model of asthma induced by ovalbumin reduced both hyperplasia and hypertrophy, reducing the thickness of the bronchial smooth muscle layer [124] (Figure 2). Furthermore, a CRNDE knockdown was able to reverse the upregulation of NE activity in the lungs of asthmatic mice, strengthening its involvement in tissue injury.

While studies and knowledge of EVs are continually expanding, it would be unfortunate to neglect the role of enucleated neutrophil cell bodies, known as cytoplasts. A major study in *Science Immunology* identified cytoplasts as key actors in neutrophilic inflammation in severe asthma [125]. Cytoplasts are derived from NET formation and exhibit functions identical to those of their parent neutrophils (e.g., chemokinesis and phagocytosis). In a murine model of allergic lung inflammation, Krishnamoorthy and coworkers established that neutrophil-derived cytoplasts were present in the lungs and mediastinal lymph nodes [125]. In vitro, cytoplasts were able to educate dendritic cells, which in turn induced T helper 17 differentiation from naïve CD4^+^ T cells. In BALF from asthmatic patients, cytoplasts and IL-17 levels were positively correlated, leading to the postulate that cytoplasts can mediate the transition between innate and adaptive immune responses through its action on Th17 differentiation [125]. More studies are absolutely required to define the role of cytoplasts in the pathogenesis of a broader range of diseases and to distinguish it from the role of nEVs.

## 4. nEVs and Their Therapeutic Potentials

To overcome the poor bioavailability and low therapeutic effects of many therapeutic molecules, researchers have undertaken efforts to develop delivery systems based on synthetic cationic polymers, lipid nanoparticles, or modified viruses. These carriers could have the advantages of being more stable than usual drugs with minimal systemic side effects. However, these classes of delivery vehicles can exhibit a lack of specificity for targeted sites and preferentially accumulate in highly vascularized tissues [126]. Moreover, viral gene delivery systems could elicit an undesirable host response. By contrast, EVs exhibit similar characteristics to the membranes of their original cells and thus present low toxicity and immunogenicity. EVs have a high stability in circulation and efficiently cross biological barriers, allowing the packaged molecules to have a high bioavailability by penetrating deeply into targeted tissues.

Over the last few years, significant progress has been realized to accentuate the efficient delivery of therapy via EVs to specific organs, tissues, and cells. The biodistribution of EVs can be modulated by engineering the surfaces of EVs to allow the expression of a wide range of targeting moieties. Strategies for inducing the targeted delivery of therapeutic EVs rely essentially on chemical modifications or genetic engineering (for reviews, see [127,128]. Chemical modification based on noncovalent methods such as receptor–ligand binding, electrostatic interaction, and hydrophobic insertion allow EVs to display a large panel of bioactive molecules [129,130]. Covalent reactions by “click chemistry” or bioconjugation can also be used to stably modify the surfaces of EVs [131,132]. Genetic engineering consists of fusing genes expressing a targeting moiety (e.g., peptides and antibodies) with a selected EV membrane protein such as tetraspanins or Lamp2b. Subsequently, donor cells transfected with these plasmids generate EVs through natural EV biogenesis that express targeting ligands on their surfaces [127,128,133]. 

Since it has been demonstrated that EVs contain and can transfer genetic material and bioactive molecules from their host cell lineage, EVs with neutrophil features would be desirable in the nanotheranostics field, which is associated with autoimmune disorders and cancer. In this sense, several studies putting the spotlight on nEVs as drug delivery systems are beginning to thrive. nEVs generated by nitrogen cavitation from human neutrophil membranes were packaged with ceftazidime, an antibiotic belonging to the class of cephalosporins, and/or with the proresolving lipid mediator resolvin D1 (RvD1) on the surface. RvD1 is able to bind to G-protein-coupled receptors and inhibits downstream NF-κB signaling pathways, triggering an increase in the phagocytic functions and apoptosis of leukocytes [134,135,136].

Previously, it has been reported that RvD was present in nEVs, which regulated macrophage efferocytosis, corresponding not only to an increase in apoptotic cell uptake but also to an enhancement of proresolving mediator biosynthesis by macrophages [55].

Therefore, the role of RvD in the acceleration of inflammation resolution makes it a molecule of choice in therapy. Under an LPS challenge, nEVs loaded with RvD1 and/or ceftazidime showed the potential to accumulate and adhere to the inflamed lung endothelium through interactions between the highly expressed integrin β_2_ on nEVs and the ICAM-1 on endothelial cells [137]. Therefore, these EVs could specifically target the inflamed endothelium and improve the treatment of lung infections. In support of this assumption, the delivery of RvD1 by intravenous injection to inflammatory sites in a *Pseudomonas aeruginosa* mouse lung infection model counteracted cytokine release and neutrophil lung infiltration. Ceftazidime transferred via nEVs could block bacterial growth and cytokine production in the lungs. The codelivery of RvD1 and ceftazidime drastically prevented bacterial proliferation and alleviated inflammation [137]. Similar results were obtained in a mouse model of *P. aeruginosa*-induced peritonitis [138]. These results could encourage further research into targeting bacterial infections and the resulting inflammation with EVs loaded with bioactive molecules.

Using a similar approach, the protective role of RvD2 in brain damage during an ischemic stroke has been shown in a middle cerebral artery occlusion mouse model developed to mimic ischemic stroke [139]. Neutrophils are the first circulating immune cells to enter the brain after an ischemic stroke, and EVs released from neutrophils can easily cross the blood–brain barrier. Based on this, RvD2-loaded nEVs were exploited to treat neuroinflammation and preserve the brain from damage following an ischemic stroke. RvD2 loaded into nEVs strikingly reduced the presence of MPO in the brain, supporting a decline in neutrophil infiltration in the mouse brain after reperfusion during ischemic stroke. Furthermore, RvD2 inhibited ICAM-1 expression in the damaged brain, favoring decreased neutrophil–endothelium interactions. In addition, RvD2 was able to prevent neuroinflammation and protect the brain from injury during the ischemia/reperfusion that occurred following an ischemic stroke, probably by decreasing the release of TNF-α, IL-6, and IL-1β [139]. Taken together, the current data on RvD1 or RvD2 loaded into nEVs and their role in regulating key immune functions could become a promising therapeutic avenue.

nEV-based drug-delivery systems provide an opportunity to improve patient outcomes in traditional ischemic stroke therapy.

This concept of neutrophil-derived drug delivery systems paves the way for the development of innovative methods in the context of personalized nanomedicine to treat a range of inflammation-related diseases such as rheumatoid arthritis (RA).

A recent paper by Headland et al. [140] brought the first evidence that nEVs are abundant in the synovial fluid of RA patients and could restore tissue homeostasis through their chondroprotective effects, preventing cartilage breakdown during RA [140]. The authors showed that mice deficient in TMEM16F, a Ca^2+^-dependent phospholipid scramblase, during phosphatidylserine exposure and microvesiculation, showed a reduction in nEVs associated with aggravated cartilage damage when these mice were subjected to K/BxN serum transfer arthritis. Purified human EVs from TNF-α-treated neutrophils displayed a rich expression of Annexin A1 (AnxA1), a proresolving protein with tissue repair properties. In vitro, these nEVs were used as models of nEVs from RA synovial fluid and were able to promote anabolic gene expression in human primary chondrocytes, preventing ECM degradation and ultimately protecting cartilage.

The protective role of nEVs through AnxA1 binding with the formyl peptide receptor 2 (FPR2/ALX) present on chondrocytes was confirmed in vivo after the intra-articular injection of AnxA1-positive nEVs in a mouse model of K/BxN arthritis. Neutrophils recruited to the joint during an arthritis flare-up release high numbers of EVs, which are able to successfully penetrate cartilage and deliver AnxA1 to the chondrocytes through the extracellular matrix. These nEVs could induce TGF-β production by chondrocytes while protecting them from apoptosis and chondroprotection in inflammation-induced cartilage damage [140].

Congruent with these observations, Zhang and coworkers [141] demonstrated that nEVs synthesized from activated human peripheral blood neutrophils can neutralize proinflammatory cytokines; inhibit proarthritogenic factors, thereby inhibiting synovial inflammation; and ameliorate joint destruction in arthritic mice [141]. In addition, joint macrophages subjected to nEVs could polarize towards a more anti-inflammatory phenotype in relation to the expression of phosphatidylserine and annexin A1 in nEVs [68,138]. nEVs could also disturb the crosstalk between macrophages and fibroblast-like synoviocytes, counteracting an excessive activation of these cells [67].

More recently, an attractive strategy has been developed based on neutrophil-derived nEVs functionalized with sub-5 nm ultrasmall Prussian blue nanoparticles (PBNPs) via click chemistry. As reported for PBNPs, PBNP-nEVs displayed catalytic antioxidant activities, effectively scavenging free radicals [142]. Along the same line, PBNP-nEVs showed the capacity to reduce the expression of NOX2, the NADPH oxidase isoform responsible for ROS production by neutrophils [142]. Collectively, these results support the fact that PBNP-nEVs are able to modulate oxidative stress in an RA microenvironment and protect cells against oxidative stress. Moreover, in an RA in vitro model, PBNP-nEVs disturbed the regulation of the PI3K/AKT, NF-κB, and mTOR signaling pathways, supporting the assumption that PBNP-nEVs could exert anti-inflammatory protection on joints by reducing cytokine-induced cell apoptosis [142]. The confirmation of this model was achieved in vivo using a mouse collagen-induced arthritis model where an intravenous injection of PBNP-nEVs reduced ankle and foot swelling. The amelioration of synovial inflammation and cartilage degeneration was associated with the neutralization of proinflammatory cytokine (TNF-α and IL-1β) production and the regulation of the T helper 17/regulatory T cell balance [142].

The possibility to use neutrophil-derived EVs in cancer therapy has recently been put forward since nEVs have been shown in vitro and in vivo to exert a notable antitumor effect by promoting tumor cell apoptosis through the activation of the caspase signaling pathway [68]. Based on this encouraging result, the authors successfully developed a method to allow targeted cancer therapy: nEVs loaded with doxorubicin and decorated with super-paramagnetic iron oxide nanoparticles ensured an appropriate targeted drug delivery and alleviated tumor growth in a mouse xenograft tumor model [68].

While the packaging of chemotherapeutics in nEVs is becoming more concrete, similar approaches using miRNAs remain to be explored. Encouraging data support the fact that miRNA-nEVs could also constitute an efficient tool in therapy, as outlined before.

In this sense, in the framework of systemic autoinflammatory diseases, miRNA-223 transferred by nEVs has been reported to have an anti-inflammatory role by exerting a synergistic effect on neutrophils and macrophages. Indeed, miRNA-223 could impede IL-18 production by macrophages by targeting the NLRP3 inflammasome. The decline in IL-18 levels had repercussions on Ca^2+^ influx, which was accompanied not only by a decrease in AKT activation and mitochondrial ROS production but also by the formation of fewer NETs. NETs have been reported to be enriched in oxidized mitochondrial DNA, which is highly proinflammatory. In compensation, NETs can induce an anti-inflammatory feedback control by inducing miRNA-223 upregulation in neutrophils through the activation of Toll-like receptor 9 [143]. Therefore, miR-223, by balancing the fine-tuned mechanism between pro- and anti-inflammatory functions, could represent a potentially attractive tool in the therapy of autoinflammatory diseases.

Despite some promising results, the engineering of therapeutic nEVs carrying miRNAs or other noncoding RNA into target tissues is at an early stage and is hindered by a number of obstacles. The functional enrichment of nEVs could be required to ensure the delivery of enough copies of the selected miRNA. In addition, given that nEVs reflect the parental cell state, a selective removal of bioactive compounds from nEVs without beneficial effects seems inevitable to avoid an amplification of the proinflammatory response and undesired side effects. Moreover, the results obtained in vitro and in vivo in animal models based on an efficient injection of carriers containing miRNAs need to reach the clinical trial level. Finally, the research on the engineering transformation and modification of nEVs for the successful administration and delivery of therapeutic nEVs to specific locations for optimal pharmacokinetics and the minimization of side effects needs to be deepened.

The use of biomimetic nEVs as drug delivery vehicles and therapeutic agents lies in the resolution of these important issues.

## 5. Conclusions

Since extracellular vesicles derived from neutrophils, based on their anti-inflammatory content, have exhibited the potential to protect tissues from injury, novel treatment strategies to attenuate inflammatory activity in the local environments can be envisioned with the aim to improve therapy in chronic diseases and cancer. nEVs can be enriched with specific RNA species, proteins, or bioactive lipids and present the assets to decrease unwanted side effects due in part to their low immunogenicity and innate stability as well as the ability to deliver therapeutic molecules to target cells with a great capacity for penetration.

For instance, an interesting possibility in the framework of nEV-inspired drug delivery systems could rely on S100A8/A9 and their dual inflammation function. In fact, besides their proinflammatory effects upon a phosphorylated state, S100A8/A9 could be oxidized and act as an oxidant scavenger with the ability to confer tissue protection from damage in an inflammatory environment [144]. Moreover, S100A8/A9 could modulate the activity of proinflammatory cytokines due to their binding affinity [145,146], avoiding the exacerbated recruitment of immune cells [147]. The described anti-inflammatory role of S100A8/A9 could reduce tissue injury by preventing an overwhelming inflammatory state provoked by uncontrolled immune cell functions. Therefore, post-translational modifications of S100A8/A9 and incorporation in nEVs with the aim to deliver them to target tissues could participate in the control of inflammation and the restoration of tissue homeostasis.

nEVs, as a delivery platform, offer a considerable scope of possibilities to shuttle appropriate bioactive molecules for dedicated action in a targeted tissue. However, knowledge of the translational potential of nEVs is required before drawing their entire benefit and exploiting them for successful application in clinics.

## Figures and Tables

**Figure 1 cells-11-03318-f001:**
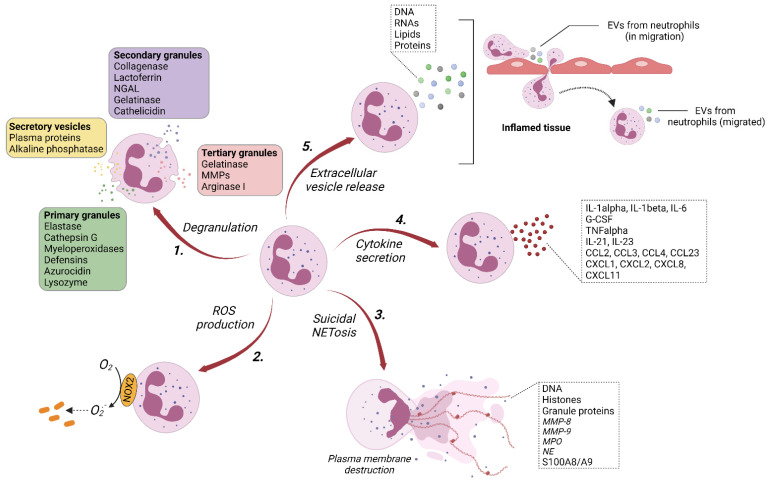
Mechanisms of neutrophil-induced inflammation. Neutrophils dispose of a large arsenal to kill pathogens. This includes **1.** degranulation, which corresponds to the release of proteolytic enzymes and cytotoxic proteins [34,35,36,37]; **2.** the production of different ROS from superoxide anions resulting from the activation of NADPH oxidase 2 (NOX2) [38,39,40]; **3.** the formation of neutrophil extracellular traps (NETs), which are associated with chromatin decondensation and the extrusion of histones, and DNA meshes decorated with granular proteins [41,42,43,44]; **4.** the secretion of a large panel of cytokines (non-exhaustive list) with proinflammatory/chemotactic/immunoregulatory functions [45,46]; and **5.** the release of extracellular vesicles loaded with cargo proteins. A dysregulation or imbalance of these different processes fosters the propagation and perpetuation of inflammation, contributing to the development of chronic diseases. All figures were created using Biorender (https://biorender.com, accessed on 21 April 2021).

**Figure 2 cells-11-03318-f002:**
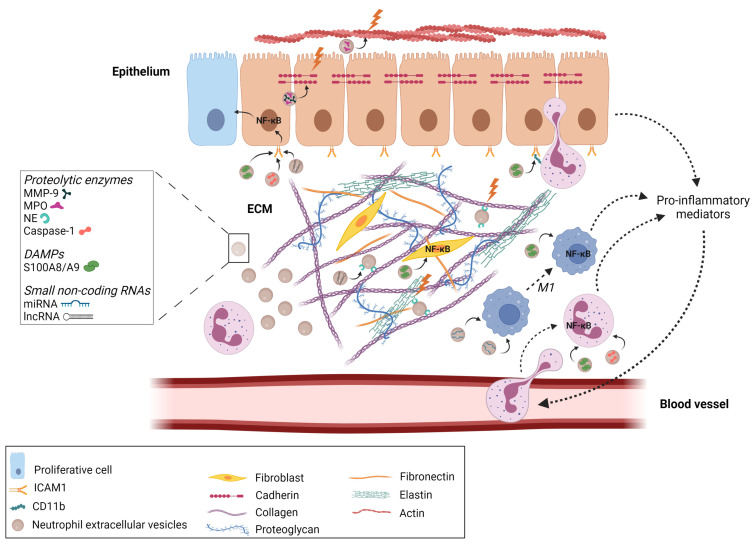
Overview of neutrophil extracellular vesicle functions involved in inflammation-inducing disease pathogenesis. Upon activation with inflammatory mediators, neutrophils rapidly release a set of extracellular vesicles into their local environment. Proteolytic enzymes (myeloperoxidases, MPO; neutrophil elastase, NE; and metalloproteinase 9, MMP-9) packaged in nEVs degrade extracellular matrix (ECM) components, disrupt the cadherin junction, and regulate actin dynamics, increasing epithelial cell permeability and facilitating immune cell invasion. Long noncoding RNA (CNRDE) can accentuate the degradation of ECM by increasing NE activity. Moreover, caspase-1 can induce the release of proinflammatory mediators by epithelial cells as well as increase ICAM-1 expression. The binding between epithelial cells and neutrophils is facilitated by S100A8/A9, which can increase the expression of CD11b on rolling neutrophils. S100A8/A9 is also able to induce NF-κB signaling in resident and infiltrated cells, resulting in an amplification of the inflammatory response through the recruitment of supplementary neutrophils. In addition, miRNAs contained in nEVs can polarize macrophages towards the proinflammatory M1 phenotype participating in the amplification loop of the inflammatory process. All figures were created using Biorender (https://biorender.com, accessed on 21 April 2021).

## Data Availability

Not applicable.

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
