# Peer review of "Neutrophil Extracellular Vesicles: A Delicate Balance between Pro-Inflammatory Responses and Anti-Inflammatory Therapies"

_cells, 2022, doi:10.3390/cells11203318_

Round 1
Reviewer 1 Report
This is a nice review of neutrophil extracellular vesicles from Zhou and Brechard. I have only a few minor suggestions to the content of the review listed below, and some minor edits for English language.
Line 112 paragraph first talking about neutrophils would be better as an introduction paragraph to section 2. Then Line 133-134 can be removed or re-written.
Figure 2. the NFkB labels are unclear in white, please change to black font
In section 4 there is comprehensive discussion of methods to target nEVs for uptake by different cell types but could the authors expand this section to discuss how they might target nEVs to specific locations within the body e.g. brain, joint, lung, tumour. How would they be delivered? How would they home from circulation to a target organ or tumour?
Minor grammatical corrections
Line 48 is there a word missing at point X? “invagination of endosomal membranes of internal multivesicular X and fusion of these compartments with the plasma membrane”
Line 57 Add words IS and AND as indicated “ Moreover, overlapping sizes of ectosomes and exosomes in such a nomenclature IS rather irrelevant AND has led to a number of confusing scientific publications”
Lines 171-173 needs re-writing. Suggest “In addition we need to keep in mind that distinct biological roles can be attributed to nEVs according to i) the nature and identity of inflammatory mediators that have induced neutrophil priming/activation ii) targeted recipient cells.
Line 238 add word IN ”… (MPO) from nEVs participate IN airway epithelial damage”…
Line 281 support should be plural “supports”
Line 296 p38 is “p38 MAPK”
Line 345 In sum should be “In summary”
CRNDE – please define is this a specific species or class of lnc-RNA?
Lines 368-370 “molecules packaged” appears twice the sentence, please re-write
Line 473 NLPRP3 should be NLRP3
Line 475 “…lesser formation of fewer NETs” is a double negative, you don’t need the word lesser please remove
Reviewer 2 Report
This review presents an important area of neutrophil biology. As the area has been not been well reviewed previously , this paper will be an important contribution to our understanding.
Author Response
We would like to thank the Reviewer 2 for reviewing our manuscript.
Reviewer 3 Report
The review by Yang Zhou and Sabrina Bréchard entitled “Neutrophil extracellular vesicles: a delicate balance between 2 pro-inflammatory responses and anti-inflammatory therapies” is a general summary of extracellular vesicles with minor scientific details. This has a piece of very preliminary information on the topic. There are a few major points here I am mentioning that may be helpful to authors-
1) In the abstract authors mentioned Extracellular vesicles (EVs) as particles, which is not appropriate with the knowledge available for them as membrane-bound vesicles and more info as per the International Society for Extracellular Vesicles (ISEV2018).
2) Authors unnecessarily over discussed general concept of EVs, and their heterogeneity (page 1-3), There are multiple reviews in a similar direction recently (https://www.sciencedirect.com/science/article/pii/S0092867420305602, https://pubmed.ncbi.nlm.nih.gov/34656117/, https://doi.org/10.3390/ijms22126417, etc). Thus a brief crisp summary would work.
3) There are more reviews cited, in comparison to good original research.
4) This is also visible in the information provided by the text mentioned, which is very general and lacks specific details.
5) Regarding Extracellular vesicles of neutrophils (nEVs), which is supposed to be the main topic of this review is very brief and general (only page 4, Line 135-188) and missed multiple substantial contributions in the field, for example, https://pubmed.ncbi.nlm.nih.gov/10510400/, https://pubmed.ncbi.nlm.nih.gov/23144171/ and many more.
6) On a similar line, nEVs in diseases and therapeutic sections are redundant in terms of specific scientific details about EVs, their contents, specific conditions, functions, etc
7) Overall, this review in its current form does not cover the topic “Neutrophil extracellular vesicles” and does not also specify their pro-inflammatory responses and anti-inflammatory nature.
Still, it is an interesting area of research, and authors shall put more effort to cover this substantially.
Reviewer 4 Report
The authors of the article cells-1944649 entitled “Neutrophil extracellular vesicles: a delicate balance between pro-inflammatory responses and anti-inflammatory therapies", wrote a comprehensive review on the implication of neutrophilic derived EVs in physiology and pathophysiology. However, there are a couple of comments that could improve the manuscript.
- This review has some novel points compared to other similar reviews. It would be interesting if the authors would also comment on some new arising data on neutrophilic cytoplasts (although technically not EVs, Nandini Krishnamoorthy et al. 2018, Sci Immunol) and could give more novelty to the manuscript.
- The authors have an extensive paragraph named “nEVs in diseases”. They also talk about procoagulant activity of EVs. Yet they do not include one of the few reports that demonstrate in clinical samples (human) ex vivo presence of neutrophilic microparticles in acute and remission phases of a human disorder (Kambas et al, 2014 Ann Rheum Dis). In my opinion the authors should include this report in this paragraph.
Author Response
We would like to thank the Reviewer 4 for her/his helpful input in view to improve our manuscript.
- This review has some novel points compared to other similar reviews. It would be interesting if the authors would also comment on some new arising data on neutrophilic cytoplasts (although technically not EVs, Nandini Krishnamoorthy et al. 2018, Sci Immunol) and could give more novelty to the manuscript.
As recommended by the reviewer, the role of neutrophil cytoplasts has been tackled in the revised manuscript at the end of the section 3 (lines 330-341).
- The authors have an extensive paragraph named “nEVs in diseases”. They also talk about procoagulant activity of EVs. Yet they do not include one of the few reports that demonstrate in clinical samples (human) ex vivo presence of neutrophilic microparticles in acute and remission phases of a human disorder (Kambas et al, 2014 Ann Rheum Dis). In my opinion the authors should include this report in this paragraph.
As suggested by the reviewer, a paragraph on the role of tissue factor, released through neutrophil extracellular vesicles, in the pathogenesis of antineutrophil cytoplasmic antibody-associated vasculitis has been added in the section 3 of the revised manuscript (lines 380-393).
Round 2
Reviewer 3 Report
The authors incorporated some suggestions.Still, it would be nice to get more info on nEV, and specific original research (which is still a small section of this review).
Author Response
As requested by the reviewer, a paragraph has been added on nEVs about what we know on their characteristics, antibacterial role and pro-inflammatory/anti-inflammatory effects (see revised manuscript lines 140-165).
Round 3
Reviewer 3 Report
The authors provided sufficient details against queries and MS is suitable for publication.